# Exclusive Breastfeeding Rates and Associated Factors in 13 “Economic Community of West African States” (ECOWAS) Countries

**DOI:** 10.3390/nu11123007

**Published:** 2019-12-09

**Authors:** Kingsley Emwinyore Agho, Osita Kingsley Ezeh, Pramesh Raj Ghimire, Osuagwu Levi Uchechukwu, Garry John Stevens, Wadad Kathy Tannous, Catharine Fleming, Felix Akpojene Ogbo

**Affiliations:** 1School of Science and Health, Western Sydney University, Locked Bag 1797, Penrith, NSW 2571, Australia; ezehosita@yahoo.com (O.K.E.); Prameshraj@hotmail.com (P.R.G.); Catharine.Fleming@westernsydney.edu.au (C.F.); 2Translational Health Research Institute (THRI), School of Medicine, Western Sydney University, Campbelltown Campus, Locked Bag 1797, Penrith, NSW 2571, Australia; K.Tannous@westernsydney.edu.au (W.K.T.); F.Ogbo@westernsydney.edu.au (F.A.O.);; 3Diabetes, Obesity and Metabolism Translational Research Unit, Western Sydney University, Campbelltown, NSW 2560, Australia; L.osuagwu@westernsydney.edu.au; 4Humanitarian and Development Research Initiative (HADRI), School of Social sciences and Psychology, Western Sydney University, Locked Bag 1797, Penrith, NSW 2751, Australia; G.Stevens@westernsydney.edu.au; 5School of Business, Western Sydney University, Locked Bag 1797, Penrith, NSW 2751, Australia; 6General Practice Unit, Prescot Specialist Medical Centre, Welfare Quarters, Makurdi, Benue State 972261, Nigeria

**Keywords:** breastfeeding, Africa, antenatal care, infants, ECOWAS, mortality

## Abstract

Exclusive breastfeeding (EBF) has important protective effects on child survival and also increases the growth and development of infants. This paper examined EBF rates and associated factors in 13 “Economic Community of West African States” (ECOWAS) countries. A weighted sample of 19,735 infants from the recent Demographic and Health Survey dataset in ECOWAS countries for the period of 2010–2018 was used. Survey logistic regression analyses that adjusted for clustering and sampling weights were used to determine the factors associated with EBF. In ECOWAS countries, EBF rates for infants 6 months or younger ranged from 13.0% in Côte d’Ivoire to 58.0% in Togo. EBF decreased significantly by 33% as the infant age (in months) increased. Multivariate analyses revealed that mothers with at least primary education, older mothers (35–49 years), and those who lived in rural areas were significantly more likely to engage in EBF. Mothers who made four or more antenatal visits (ANC) were significantly more likely to exclusively breastfeed their babies compared to those who had no ANC visits. Our study shows that EBF rates are still suboptimal in most ECOWAS countries. EBF policy interventions in ECOWAS countries should target mothers with no schooling and those who do not attend ANC. Higher rates of EBF are likely to decrease the burden of infant morbidity and mortality in ECOWAS countries due to non-exposure to contaminated water or other liquids.

## 1. Introduction

Exclusive breastfeeding (EBF) is the best source of nutrients for the healthy growth and development of newborns, as well as a natural immunity for protection against infectious (e.g., diarrhoea) [1,2,3,4] and long-term chronic diseases [4,5]. The World Health Organization and United Nations Children’s Fund (WHO/UNICEF) recommends that nursing mothers should practice EBF, defined as providing the infant human breastmilk only, and oral rehydration solution, or drops/syrups of vitamins, minerals, or medicines, when required [6]. This is because EBF is strongly correlated with a reduced burden of infant and child morbidity and mortality [7]. A recently published report on EBF indicated that approximately 12% of deaths among children under five years old could be averted annually in low- and middle-income countries if all neonates were exclusively breastfed [8]. In sub-Saharan Africa, including Economic Community of West African States (ECOWAS), approximately 42% of diarrhoea-related deaths among children younger than five years of age could be attributed to prelacteal foods or unimproved water or liquids provided to newborns [9]. Mothers who exclusively breastfeed also have a reduced risk of developing type 2 diabetes mellitus and breast and ovarian cancers [4]. ECOWAS is a political and economic union of fifteen member countries in West Africa. The members include Benin, Burkina Faso, Cabo Verde, Côte d’Ivoire, The Gambia, Ghana, Guinea, Guinea-Bissau, Liberia, Mali, Niger, Nigeria, Senegal, Sierra Leone, and Togo. ECOWAS main aim is to promote socioeconomic integration among member countries in order to improve living standards and health outcomes of the population and promote the economic growth of member states [10].

Despite the numerous evidence showing the benefits of EBF for the mother–infant pair [1,4,11], EBF has remained low across ECOWAS countries and varies widely between and within countries [12,13,14,15]. Results from the recent Demographic and Health Survey (DHS) in ECOWAS countries between 2010 and 2018 revealed that many countries in the region had EBF rates that were well below the global target (50%) required to considerably reduce infant mortality [16,17]. This low rate of EBF indicates that a substantial proportion of newborns in ECOWAS countries are fed with water and/or prelacteal foods before the age of at least six months. This will, in turn, predispose the infant to gastrointestinal infections due to intake of contaminated prelacteal [18] or complementary foods [1,3,19]. In a recent study conducted in Nigeria, one of the ECOWAS countries with the largest population and economy [10], an estimated 22,371 diarrhoea-related deaths among children under five years could have been averted in 2016 if nursing mothers had strictly adhered to appropriate breastfeeding practices, particularly EBF [20].

In ECOWAS countries (Nigeria, Ghana, Mali, and Niger), past studies conducted based on data collected between 2006 and 2012 have elucidated some factors associated with appropriate EBF practice. The factors include place of residence and birthing [12], higher maternal and paternal education [14,21,22,23], higher household wealth index [21,23], child’s birth order, frequent (≥4) antenatal and postnatal visits [12,21,23], average size of newborns, vaginal birthing [12,24], assistance during delivery, maternal employment, and higher maternal age [12,14,21]. However, none of the previous studies had investigated EBF rates across ECOWAS countries or identified what acts as an enabler or barrier to appropriate EBF behaviours using the most recent country-specific and standardised data. Additionally, some of the previous studies were conducted at the subnational level in Nigeria and Ghana, where their findings may not inform national-level policy interventions [15,25]. Some of the previous nationwide studies used old datasets, which may not reflect the current sociodemographic and economic settings of those countries [12,23].

Understanding the current EBF rates and associated factors across ECOWAS countries is essential to key stakeholders in formulating integrated and effective health policy interventions. Furthermore, the study findings could offer credible information needed to prioritise cost-effective, evidence-based interventions that may rapidly improve EBF rates and subsequently reduce the high burden of under-five deaths in ECOWAS countries [26]. In the current United Nations Decade of Action on Nutrition (2016–2025) [16], the present study will provide policy-relevant data about EBF practice that would assist ministries of health, international agencies, and nongovernmental organizations to design programs that promote, protect, and support appropriate EBF practice in the region. This study aimed to examine EBF rates and the potential demographic, socioeconomic, and proximate factors that are associated with EBF in 13 ECOWAS countries using the combined dataset from 2010 to 2018.

## 2. Materials and Methods

The analyses were based on the most recent DHS dataset from 13 ECOWAS countries, which were obtained from a password-enabled Measure DHS website [27]. The DHS data were nationally representative and population-based surveys, collected by country-specific ministries of health or other relevant government-owned agencies, with technical support largely provided by Inner City Fund (ICF) International. These surveys were comparable, given the standardised nature of the data collection methods and instruments [28].

The DHS collects demographic data and population health status of people, including reproductive health, maternal and child health, mortality, nutrition, and self-reported health behaviour among adults [28]. Information was collected from eligible women, that is, all women aged 15–49 years who were either permanent residents in the households or visitors present in the households on the night before the survey. Child health information was collected from the mother based on the youngest child aged less than five years, with response rates that ranged from 96% to 99% [27]. Detailed information on the sampling design and questionnaire used is provided in the respective country-specific Measure DHS reports [27]. Our analyses were restricted to the last born child aged 0–5 months and living with the respondent, which yielded a weighted total of 19,735 infants for all 13 ECOWAS countries.

### 2.1. Outcome, Confounding and Exploratory Variables

EBF rate was estimated using the WHO/UNICEF definitions for assessing infant and young feeding practices in populations [29] and used by Measure DHS. EBF was measured as the proportion of infants 0–5 months of age who were fed exclusively with breast milk (but allows oral rehydration solution and drops or syrups of vitamins and medicines when required). Information on EBF was collected based on maternal recall on feeds provided to the infant in the last 24 h. EBF was categorized as “Yes” (1 = if the infant was exclusively breastfed) or “No” (0 = if the infant was not exclusively breastfed).

Previous studies conducted in sub-Saharan African countries that examined factors associated with EBF [23,30,31,32,33] played a vital role in determining the potential confounding variables for this study. The confounding variables were subdivided into four groups, and these were country and demographic factors, socioeconomic factors, access to media factors, and healthcare utilisation factors. The country variables were Benin, Burkina Faso, Côte d’Ivoire, The Gambia, Ghana, Guinea, Liberia, Mali, Niger, Nigeria, Senegal, Sierra Leone, and Togo. We considered Benin as the referenced category because it was the first country on the list of ECOWAS countries. The demographic variables were place of residence (urban or rural), mother’s age, marital status, combined birth rank (the position of the youngest under-five child in the family), and birth interval (the interval between births; that is, whether there were no previous births, birth less 24 months prior, or birth more than or equal to 24 months prior), sex of baby, age of the child, and perceived size of the newborn by the mother. The socioeconomic level factors considered were maternal education, maternal work status, maternal literacy, and household wealth index variable. For the combined datasets, the household wealth index was constructed using the “hv271” variable. In the household wealth index categories, the bottom 20% of households was arbitrarily referred to as the poorest households, and the top 20% as the richest households, and was divided into poorest, poor, middle, rich, and richest. Access to media factors consists of the frequency of mothers listening to the radio, watching television, and reading newspapers or magazines. Healthcare utilisation factors were considered and included (birthplace, birth order, mode of delivery, delivery assistance, and antenatal clinic visits (ANC).

### 2.2. Statistical Analysis

Population-level weights were used for survey tabulation, which adjusts for a unique country-specific stratum, and clustering was used to determine the percentage, frequency count, and univariate and multivariate logistic regression of all selected characteristics. Country-specific weights were used for the Taylor series linearization method in the surveys when estimating 95% confidence intervals around the rate of EBF in each country.

For the combined dataset, sampling weight was denormalised, and a new population-level weight was created by dividing the sampling weights by the denormalised weight. We then created a unique country-specific cluster and strata because each country had individual clusters and strata in the DHS. This was done to account for the uneven country-specific population across the organisation and to avoid the effect of countries with a large population (such as Nigeria with over 175 million people in 2013) offsetting countries with a small population (such as The Gambia with about 1.8 million people in 2013) [10].

In the multivariate analyses, the factors associated were further tested by adjusted odds ratios (AOR) using hierarchical multiple logistic regression analyses as described in Table 1. The first stage (Model 1) included country and demographic factors. The second stage (Model 2) also included socioeconomic factors. The third stage (Model 3) added access to media covariates. The fourth and final stage (Model 4) added healthcare utilisation factors. The objective of this modelling strategy was to allow for a comparison of the relationship between each of the different sets of covariates in examining factors associated with EBF. All analyses were performed in Stata version 14.0 (Stata Corp, College Station, Texas, USA).

## 3. Results

### 3.1. Characteristics of the Sample and Unadjusted Analyses for EBF

The overall EBF rate for all infants aged 0–5 months in the 13 ECOWAS countries was 31.0% between 2010 and 2018 (Figure 1). Higher EBF rates in ECOWAS countries ranged from 52.0% in Ghana to 58.0% in Togo, while Nigeria and Côte d’Ivoire had the lowest EBF rates of less than 20%. Table 2 illustrates the characteristics of the sample of infants aged 0–5 months and their unadjusted odds ratios. Overall, the number of mother–infant dyads included in our sample varied by country, with Nigeria contributing the most (20.0% of the study sample) and Togo the least (3.3%). Bivariate analyses revealed that Nigerian mothers were less likely to exclusively breastfeed their babies compared to those who resided in Benin (OR = 0.29, 95% CI: 0.22, 0.38).

Infants whose mothers were from the richer and richest households were significantly more likely to exclusively breastfeed compared to those from poor or poorer households (OR = 1.51, 95% CI: 1.21, 1.88 for richer and OR = 1.96, 95% CI: 1.54, 2.51 for richest). Mothers who watched television almost every day reported higher odds of EBF compared to those who did not watch television at all (OR = 5.44, 95% CI: 3.14, 9.43). This association was also observed among mothers who listened to the radio almost every day (OR = 3.98, 95% CI: 2.41, 6.58). Approximately 57.0% of mothers who delivered their babies in health facilities were significantly more likely to exclusively breastfeed their babies compared to those 43.0% who delivered at home (OR = 1.73, 95% CI: 1.49, 2.01).

Mothers who had caesarean delivery reported much lower proportions (3.5%) of EBF compared to those who had vaginal delivery (96.2%). More than half (51.8%) of mothers had four or more ANC visits, and those mothers who attended ANC visits reported a higher likelihood of EBF compared to those who did not attend any ANC visits (OR = 2.75, 95% CI: 2.15, 3.51) (Table 2).

### 3.2. Factors Associated with Exclusive Breastfeeding in Multivariate (Adjusted) Analyses

Table 3 illustrates the factors associated with EBF among infants aged 0–5 months in multivariate analyses. Among the 13 ECOWAS countries, mothers in Côte d’Ivoire and Nigeria showed the lowest likelihood to engage in EBF practice compared to those in Benin (OR = 0.18, 95% CI: 0.12, 0.28 for Côte d’Ivoire and OR = 0.33, 95% CI: 0.24, 0.45 for Nigeria). Mothers who resided in Burkina Faso, Guinea, Niger, and Sierra Leone were also less likely to engage in EBF compared to those in Benin (Model 4; OR = 0.49, 95% CI: 0.37, 0.64 for Burkina Faso, OR = 0.47, 95% CI: 0.26, 0.84 for Guinea, OR = 0.50, 95% CI: 0.32, 0.79 for Niger, and OR = 0.62, 95% CI: 0.40, 0.97 for Sierra Leone). Liberian and Togolese mothers were more likely to practice EBF compared to their counterparts in Benin (Model 4; OR = 1.65, 95% CI: 1.03, 2.66 for Liberia and OR = 1.52, 95% CI: 1.04, 2.21 for Togo) (Table 3).

Mothers who perceived their babies to be of average birth size were more likely to engage in EBF compared to those who perceived their babies to be of small birth size (OR = 1.22, 95% CI: 1.00, 1.50). Mothers who lived in rural areas and those who had female infants reported higher odds of EBF compared to their counterparts (OR = 1.25, 95% CI: 1.01, 1.55 for rural and OR = 1.17, 95% CI: 1.00, 1.37 for female infants) (Table 3).

Higher maternal age (35–49 years) and education (primary and above education) were associated with increased odds of EBF practice compared to young maternal age (15–19 years) and no maternal education, respectively (OR = 1.50, 95% CI: 1.12, 1.99 for maternal age, and OR = 1.32, 95% CI: 1.06, 1.65 for primary and OR = 1.83, 95% CI: 1.18, 2.82 for secondary and above education). Mothers who made ANC visits had higher odds of EBF practice compared to those who did not make any ANC visits (OR = 1.52, 95% CI: 1.14, 2.03 for 1–3 ANC visits and OR = 1.66, 95% CI: 1.24, 2.23 for four or more ANC visits) (Table 3).

Increasing age of the infant was the only common factor associated with a lower likelihood of EBF in ECOWAS countries (see Table 3) and across all ECOWAS countries (see Appendix A).

## 4. Discussion

Our study indicated that the overall rate of EBF among infants aged 0–5 months of life in 13 ECOWAS countries was at a suboptimum level between 2010 and 2018 compared to the global target of 50% by 2025 [17]. There is a need for further improvement in order to gain the full benefits of EBF as 10 out of the 13 ECOWAS countries had EBF rates between 13% and 47%. This study found that place of residence (rural), maternal education (primary or secondary), mother’s age (35–49 years), and antenatal care visit (1–3 or ≥4), increased the likelihood of EBF in 13 ECOWAS countries. Baby size at birth (average) and child’s gender (female) were associated with EBF.

The present study showed that nursing mothers who resided in rural areas were more likely to exclusively breastfeed their newborns for less than six months compared with their urban counterparts. This finding is consistent with previous studies conducted in Ethiopia [34], Egypt [35], Malaysia [36], and Lebanon [37]. This finding, however, contradicts earlier reports from Nigeria [30] and Bangladesh [38], which indicated a negative association between rural residence and EBF. An increased likelihood of the association between EBF and rural residence noted in the current study (even after accounting for maternal employment status) could be attributed to a high number of nonworking rural women (rural: 68% vs. urban: 32%). This can, in turn, allow unlimited time for nonworking rural women to breastfed their babies exclusively. Most working nursing mothers reside in urban areas, and labour law concerning maternity leave duration in most ECOWAS countries remains a serious challenge for promotion, protection, and support of EBF practice [13,39]. Fewer than six months of maternity leave are granted to nursing mothers in many ECOWAS countries; for example, in Ghana (12 weeks) [40], Nigeria (16 weeks) [39], and Senegal (14 weeks) [41]. This often leads to early weaning of EBF among working nursing mothers. In addition to the short duration of maternity leave, inadequate facilities and unallocated official time for breastfeeding in the workplace also limit EBF [42]. Urban regions (e.g., urban slums) with potentially disadvantaged communities may deserve special focus with targeted interventions for improving EBF practice in these ECOWAS countries.

The odds of EBF practice were higher among nursing mothers who had primary or higher education, in comparison to their counterparts who had no formal education. This is contrary to previous studies conducted in Ethiopia [34] and Bangladesh [38], where mothers who had no formal education were more likely to report EBF practice. Nevertheless, other studies conducted in sub-Saharan African countries [21,23,33] have also suggested that higher maternal education was significantly associated with EBF practice. Educated mothers are more knowledgeable in utilizing maternal health care services appropriately, including ANC [43], postnatal care (PNC) [44], and institutional delivery [45], which remains an important initiative to scale up EBF participation [46]. To add, past studies have suggested that no formal education among mothers was associated with both underutilization of ANC [47,48] and PNC [49] services.

Furthermore, older mothers (≥35 years) reported a higher likelihood of EBF practice compared with younger mothers. This result is not consistent with previous studies [35,36,50], which indicated that younger mothers were more likely to maintain EBF practice than older mothers. The difference in outcomes may be linked to maternal age classification, characteristics adjusted for, and diverse populations. However, our finding is in line with previous studies conducted in Bangladesh [38] and six low- and middle-income countries [51] that used similar data and maternal age limits. An explanation for the higher odds of EBF noted among older mothers may be their experience in child-rearing, better knowledge of breastfeeding, and being well informed of the benefits of breastfeeding to both newborns and mothers. Additionally, older mothers may also have more time for EBF due to decreased paid job opportunities compared with younger mothers.

Mothers who had one or more ANC visits to the health institution prior to childbirth showed higher odds of EBF practice compared to those who did not attend any ANC visits. This finding is similar to cross-sectional studies carried out in sub-Saharan Africa [23,30] and Bangladesh [38]. The increased odds of EBF in the current study may be attributed to breastfeeding counselling received through the Baby-Friendly Hospital Initiative (BFHI) program during the ANC visits, which may have improved mothers’ knowledge and benefits of EBF practice. This is supported by previous evidence from regional Egypt and Ethiopia [35,52], which showed that breastfeeding counselling during ANC was positively related to EBF practice.

This study found that female infants were also associated with higher EBF rates, and this finding is similar to the research conducted in Nigeria, which found that female infants were 2.13 times more likely to be exclusively breastfed than male infants [30]. A study conducted by Johns Hopkins Children’s Center revealed that the protective effect of EBF was higher in female infants than male infants [53]. Even though our result on female babies was positively related to EBF practice, caution needs to be exercised in concluding this finding and would warrant further investigation to better understand the impact of gender on EBF in ECOWAS countries. Furthermore, our study also showed that increasing infant age was the only common factor associated with a lower likelihood of EBF in all ECOWAS countries. This finding has been reported in India, where EBF practice decreases faster with increasing infant age in Indian regions with higher socioeconomic status women [54]. Country-specific decline in EBF associated with increasing infant age has not been well studied in ECOWAS countries, and thus, would need further assessment to inform targeted interventions. Reasons why other factors were not significant across the ECOWAS countries varied, including methodological (e.g., sample size) and/or qualitative factors (e.g., cultural norms).

Babies perceived by their mothers to be of average size at birth had an increased likelihood of EBF practice compared to babies perceived by their mothers to be of small size at birth. This finding was similar to a cross-sectional study carried out in Timor-Leste in 2014, which showed that babies perceived to be of a small size at birth by their mothers were less likely to be EBF [55]. Furthermore, a cross-sectional study done in the United States in 2011 reported that very low birth weight (a proxy for babies perceived to be of a small size at birth) was significantly related to a lower likelihood of EBF practice compared to babies born at normal or above-normal birth weights [56]. The significantly lower chance of EBF practice for newborns perceived to be of a small size by their mothers in ECOWAS countries may be attributed to obstacles related to breastfeeding small-sized babies, such as poor sucking and illness (e.g., hypoglycaemia), leading to babies being kept away from their mothers [57], resulting in early introduction of complementary foods and drinks. Although caesarean birthing was associated with non-EBF practice in the bivariate analyses, this association was not significant in multivariate analyses. The possible reasons why caesarean birth was related to non-EBF have been described elsewhere [12,24,58], including a limited number of BFHI-certified maternal health centres, or health practitioners’ limited knowledge of appropriate EBF immediately post-birth [59,60]. The BFHI was introduced in 1990 by WHO/UNICEF to promote, protect, and support breastfeeding [61] and has been shown to be successful in increasing EBF participation worldwide [62,63].

The main strengths of our study were the nationally representative sample [27], the comprehensive data on standard infant feeding indicators, and the appropriate adjustments for sampling design in the analysis. There are five limitations of note in this study: Firstly, the subjectivity of information in recalling the time of EBF, especially among mothers who have older infants. Secondly, measurement error and recall bias by mothers may have underestimated or overestimated our effect sizes because information concerning some of the study variables was obtained from mothers whose youngest child may have been up to five years of age. Thirdly, findings in the study cannot be regarded as causal because they were obtained with a cross-sectional design. Fourthly, the study was unable to account for other study and confounding factors, including information on the health status of mothers, partner support, smoking status and alcohol intake, and prenatal breastfeeding intention as documented elsewhere [59,60]. Lastly, the 24 h recall approach was used for EBF investigation, and it is possible that day-to-day variability in nutritional intake may have affected EBF estimates.

## 5. Conclusions

In summary, the rates of EBF in 10 out of the 13 ECOWAS countries were below the WHO global nutrition target of 50% for breastfeeding. The low rates of EBF in ECOWAS countries may be contributing to higher infant and child morbidity and mortality in each country, given that it is a strong indicator for infant and child growth and survival. At the community level, “baby-friendly” initiative needs to be implemented as a trial in order to empower the community for better nutrition and health improvement of children, as well as to test its effectiveness. At the individual level, the practice of giving water and other liquids is a significant contributor to the decrease in EBF in Africa [18,64]. Awareness must be spread to inform mothers that their babies do not require water and liquids until the completion of the recommended time of six months. These are crucial messages that could be spread by community-level campaigns, including peer support community intervention for young mothers who are vulnerable and have lower rates of EBF. However, further improvement is needed in order to gain the full benefits of breastfeeding practices. Our results suggest that breastfeeding interventions should target urban residents, working mothers, mothers who do attend antennal care, those who give birth to male infants, and mothers who perceive their babies to be small at birth.

## Figures and Tables

**Figure 1 nutrients-11-03007-f001:**
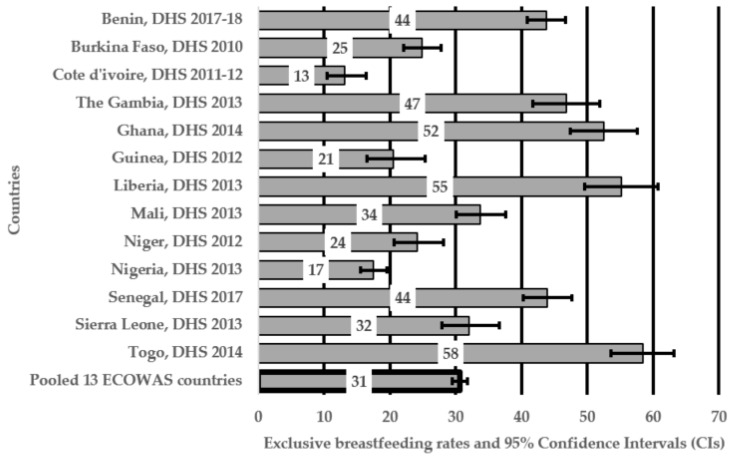
Exclusive breastfeeding rates of infants aged 0–5 months and corresponding 95% confidence intervals in 13 “Economic Community of West African States” (ECOWAS) countries.

**Table 1 nutrients-11-03007-t001:** Potential covariates used for hierarchical survey logistic regression model.

Model 1	Model 2	Model 3	Model 4
Country ^1^	Country ^1^	Country ^1^	Country ^1^
Demographic	Demographic	Demographic	Demographic
Residence	Socioeconomic	Socioeconomic	Socioeconomic
Mother’s age	Household Wealth Index	Access to media	Access to media
Marital status	Work in the last 12 months	Frequency of reading newspaper or magazine	Healthcare utilization factors
Birth rank and birth interval	Maternal education	Frequency of listening to Radio	Place of delivery
Sex of baby	Maternal Literacy	Frequency of watching Television	Mode of delivery
Age of child			Type of delivery assistance
Size of baby			Antenatal Clinic visits

^1^ Benin, Burkina Faso, Côte d’Ivoire, The Gambia, Ghana, Guinea, Liberia, Mali, Niger, Nigeria, Senegal, Sierra Leone or Togo.

**Table 2 nutrients-11-03007-t002:** Individual, household, and community level characteristics and unadjusted odd ratios (OR) (95% CI) of exclusive breastfeeding (EBF) among infants aged 0–5 months in 13 ECOWAS countries.

Characteristic	*n* *	% *	%	EBF			
				OR	95% CI		*p*-Value
**Demographic Factors**							
**Country**							
Benin	1475	7.5	8.9	1.00			
Burkina Faso	1837	9.3	10.0	0.46	0.37	0.58	<0.001
Côte d’Ivoire	1110	5.6	5.3	0.17	0.12	0.26	<0.001
The Gambia	1422	7.2	6.6	0.99	0.64	1.52	0.946
Ghana	805	4.1	4.2	1.33	0.98	1.80	0.069
Guinea	981	5.0	4.9	0.42	0.26	0.68	<0.001
Liberia	927	4.7	4.9	1.50	1.02	2.21	0.040
Mali	1192	6.0	6.9	0.66	0.51	0.86	0.002
Niger	2196	11.1	9.0	0.46	0.32	0.64	<0.001
Nigeria	3996	20.2	20.2	0.29	0.22	0.38	<0.001
Senegal	1298	6.6	7.3	1.09	0.84	1.41	0.537
Sierra Leone	1842	9.3	7.7	0.55	0.39	0.78	0.001
Togo	655	3.3	4.2	1.81	1.37	2.39	<0.001
**Residence**							
Urban	6630	33.6	29.8	1.00			
Rural	13,106	66.4	70.2	0.79	0.66	0.94	0.008
**Mother’s age (in years)**							
15–19	7215	36.6	36.3	1.00			
20–34	9267	47.0	46.0	1.01	0.88	1.17	0.876
35–49	3253	16.5	17.7	1.00	0.84	1.20	0.968
**Marital status**							
Currently married	18,357	93.0	93.3	1.00			
Formerly married ^	275	1.4	1.5	0.80	0.52	1.23	0.314
Never married	1103	5.6	5.1	1.10	0.79	1.53	0.574
**Child Age (in months)**	-	-	-	0.70	0.67	0.73	<0.001
**Birth order**							
First-born	3913	19.8	20.0	1.00			
2nd-4th	9579	48.5	46.6	0.93	0.79	1.09	0.355
5 or more	6243	31.6	33.4	0.70	0.59	0.83	<0.001
**Preceding birth interval (*n* = 19,695)**							
No previous birth	3913	19.8	19.7	1.00			
<24 months	1831	9.3	9.2	0.85	0.65	1.11	0.243
≥24 months	13,951	70.7	70.9	0.83	0.72	0.97	0.016
**Combined Birth rank and birth interval**							
1st birth rank	3913	19.8	19.7	1.00			
2nd/3rd birth rank, more than 2 years interval	8191	41.5	40.1	0.91	0.77	1.08	0.281
2nd/3rd birth rank, less than or equal to 2 years interval	1388	7.0	6.7	1.00	0.76	1.33	0.963
4th birth rank, more than 2 years interval	5432	27.5	29.2	0.71	0.60	0.85	<0.001
4th birth rank, less than or equal to 2 years interval	811	4.1	4.3	0.66	0.47	0.93	0.016
**Sex of baby**							
Male	9777	49.5	50.1	1.00			
Female	9958	50.5	49.9	1.09	0.95	1.24	0.206
**Size of the baby (*n* = 19,584)**							
Small	3701	18.8	18.5	1.00			
Average	8433	42.7	42.7	1.21	1.02	1.42	0.025
Large	7450	37.8	37.8	1.13	0.96	1.33	0.155
**Socioeconomic factors**							
**Household Wealth Index**							
Poorest	3252	16.5	20.2	1.00			
Poorer	3771	19.1	20.2	1.43	1.13	1.81	0.003
Middle	4071	20.6	20.4	1.45	1.16	1.82	0.001
Richer	3790	19.2	19.5	1.51	1.21	1.88	<0.001
Richest	4851	24.6	19.7	1.96	1.54	2.51	<0.001
**Work in the last 12 months (*n* = 19,733)**							
Not working	9252	46.9	44.2	1.00			
Working	10,481	53.1	55.8	0.92	0.80	1.07	0.288
**Maternal education (*n* = 19,732)**							
No formal education	12105	61.3	61.9	1.00			
Primary	3251	16.5	17.9	1.36	1.14	1.63	0.001
Secondary and above	4376	22.2	20.2	1.86	1.57	2.21	<0.001
**Maternal Literacy (*n* = 19,603)**							
Cannot read at all	14,935	75.7	77.6	1.00			
Able to read only part of sentences	4668	23.7	21.8	1.56	1.33	1.82	<0.001
**Access to media**							
**Frequency of reading a newspaper or magazine (*n* = 19,688)**							
Not at all	17,770	90.0	91.6	1.00			
Less than once a week	998	5.1	4.4	1.63	1.22	2.19	0.001
At least once a week/almost every day	914	4.6	3.7	1.45	1.03	2.05	0.035
**Frequency of listening to radio (*n* = 19,703)**							
Not at all	6844	34.7	36.2	1.00			
Less than once a week	4957	25.1	25.2	1.18	1.00	1.41	0.054
At least once a week/almost every day	7734	39.2	37.6	1.35	1.16	1.58	<0.001
**Frequency of watching television (*n* = 19,684)**							
Not at all	11,558	58.6	61.0	1.00			
Less than once a week	2692	13.6	14.6	1.18	0.98	1.43	0.075
At least once a week/almost every day	5286	26.8	23.5	1.36	1.14	1.63	0.001
**Healthcare utilization factors**							
**Place of delivery**							
Home	8493	43.0	42.0	1.00			
Health facility	11,242	57.0	58.0	1.73	1.49	2.01	<0.001
**Mode of delivery (*n* = 19,672)**							
Noncaesarean	18,990	96.2	96.4	1.00			
Caesarean section	682	3.5	3.4	1.54	1.11	2.15	0.010
**Type of delivery assistance (*n* = 15,940)**							
Health professional	12,342	65.0	62.0	1.00			
Traditional birth attendant	476	2.4	2.7	0.65	0.48	0.89	0.007
Other untrained	1913	9.0	10.9	0.84	0.63	1.13	0.251
No one	1209	6.1	6.0	0.50	0.39	0.65	<0.001
**Antenatal Clinic visits (*n* = 19,147)**							
None	2446	12.4	12.8	1.00			
1–3	6485	32.9	34.4	2.12	1.65	2.71	<0.001
≥4	10,216	51.8	50.4	2.75	2.15	3.51	<0.001

* Weighted sample sizes and percentages; ^ divorced/separated/widowed.

**Table 3 nutrients-11-03007-t003:** Adjusted OR (95% CI) of factors associated with exclusive breastfeeding among infants aged 0–5 months in 13 ECOWAS countries.

Characteristic	Model 1				Model 2				Model 3				Model 4			
	aOR	95% CI		*p*-Value	aOR	95% CI		*p*-Value	aOR	95% CI		*p*-Value	aOR	95% CI		*p*-Value
**Demographic factors**																
Benin	1.00				1.00				1.00				1.00			
Burkina Faso	0.46	0.36	0.58	<0.001	0.46	0.36	0.59	<0.001	0.46	0.35	0.59	<0.001	0.49	0.37	0.64	<0.001
Côte d’Ivoire	0.16	0.11	0.25	<0.001	0.17	0.11	0.26	<0.001	0.17	0.11	0.26	<0.001	0.18	0.12	0.28	<0.001
The Gambia	1.06	0.70	1.60	0.795	0.97	0.63	1.48	0.872	0.97	0.64	1.47	0.878	0.89	0.56	1.40	0.618
Ghana	1.43	1.01	2.04	0.047	1.17	0.77	1.77	0.462	1.16	0.76	1.78	0.493	1.15	0.73	1.80	0.557
Guinea	0.39	0.25	0.61	<0.001	0.40	0.24	0.65	<0.001	0.40	0.24	0.65	<0.001	0.47	0.26	0.84	0.010
Liberia	1.76	1.19	2.61	0.005	1.72	1.15	2.57	0.008	1.73	1.15	2.59	0.008	1.65	1.03	2.66	0.039
Mali	0.64	0.49	0.85	0.002	0.76	0.55	1.04	0.083	0.76	0.55	1.04	0.086	0.81	0.57	1.13	0.211
Niger	0.46	0.32	0.67	<0.001	0.45	0.31	0.66	<0.001	0.45	0.31	0.65	<0.001	0.50	0.32	0.79	0.003
Nigeria	0.28	0.22	0.37	<0.001	0.26	0.20	0.34	<0.001	0.26	0.20	0.34	<0.001	0.33	0.24	0.45	<0.001
Senegal	1.17	0.88	1.56	0.279	1.11	0.81	1.51	0.518	1.12	0.81	1.54	0.497	1.08	0.77	1.52	0.665
Sierra Leone	0.56	0.38	0.83	0.004	0.53	0.36	0.78	0.001	0.54	0.37	0.78	0.001	0.62	0.40	0.97	0.037
Togo	2.00	1.48	2.68	<0.001	1.82	1.34	2.47	<0.001	1.58	1.12	2.23	0.009	1.52	1.04	2.21	0.031
**Residence**																
Urban	1.00				1.00				1.00				1.00			
Rural	0.98	0.81	1.18	0.838	1.24	1.01	1.51	0.035	1.24	1.01	1.51	0.037	1.25	1.01	1.55	0.042
**Mother’s age**																
15–19 years	1.00				1.00				1.00				1.00			
20–34 years	1.23	1.03	1.48	0.022	1.20	1.00	1.45	0.051	1.22	1.01	1.46	0.041	1.17	0.96	1.42	0.121
35–49 years	1.50	1.15	1.95	0.003	1.46	1.12	1.90	0.005	1.47	1.13	1.92	0.004	1.50	1.12	1.99	0.006
**Marital status**																
Currently married	1.00				1.00				1.00				1.00			
Formerly married ^	0.80	0.50	1.30	0.370	0.82	0.51	1.33	0.427	0.83	0.51	1.34	0.441	0.78	0.45	1.36	0.386
Never married	0.81	0.56	1.17	0.260	0.72	0.49	1.06	0.096	0.73	0.50	1.07	0.111	0.75	0.50	1.13	0.167
**Combined Birth rank and birth interval**																
1st birth rank	1.00				1.00				1.00				1.00			
2nd/3rd birth rank, more than 2 years interval	0.88	0.73	1.06	0.186	0.96	0.78	1.17	0.688	0.95	0.78	1.17	0.651	0.93	0.75	1.16	0.542
2nd/3rd birth rank, less than or equal to 2 years interval	0.97	0.71	1.32	0.843	1.03	0.76	1.40	0.858	1.03	0.76	1.39	0.861	0.85	0.60	1.20	0.361
4th birth rank, more than 2 years interval	0.61	0.48	0.78	<0.001	0.75	0.57	0.97	0.031	0.74	057	0.97	0.030	0.76	0.56	1.03	0.074
4th birth rank, less than or equal to 2 years interval	0.64	0.44	0.93	0.019	0.75	0.50	1.01	0.146	0.74	0.50	1.10	0.141	0.81	0.52	1.28	0.358
**Sex of baby**																
Male	1.00				1.00				1.00				1.00			
Female	1.12	0.98	1.29	0.102	1.13	0.98	1.30	0.094	1.13	0.98	1.31	0.082	1.17	1.00	1.37	0.046
**Child Age (in months)**	0.67	0.64	0.69	<0.001	0.66	0.64	0.69	<0.001	0.66	0.64	0.69	<0.001	0.67	0.64	0.71	<0.001
**Size of the baby**																
Small	1.00				1.00				1.00				1.00			
Average	1.26	1.05	1.51	0.013	1.23	1.02	1.48	0.032	1.23	1.02	1.48	0.030	1.22	1.00	1.50	0.050
Large	1.22	1.01	1.47	0.039	1.18	0.98	1.43	0.081	1.19	0.98	1.43	0.078	1.18	0.95	1.45	0.135
**Socioeconomic factors**																
**Household Wealth Index**																
Poorest					1.00				1.00				1.00			
Poorer					1.39	1.04	1.84	0.024	1.39	1.04	1.85	0.026	1.34	0.99	1.80	0.056
Middle					1.35	1.04	1.75	0.025	1.35	1.04	1.76	0.026	1.13	0.86	1.50	0.369
Richer					1.32	1.03	1.68	0.027	1.31	1.02	1.68	0.035	1.14	0.87	1.48	0.340
Richest					1.46	1.09	1.95	0.010	1.44	1.05	1.99	0.025	1.20	0.86	1.68	0.272
**Work in the last 12 months**																
Not working					1.00				1.00				1.00			
Working					1.00	0.85	1.17	0.962	0.99	0.84	1.16	0.911	1.01	0.85	1.20	0.899
**Maternal education**																
No education					1.00				1.00				1.00			
Primary					1.22	0.99	1.51	0.058	1.22	0.99	1.51	0.058	1.32	1.06	1.65	0.015
Secondary and above					1.79	1.12	2.86	0.014	1.84	1.15	2.93	0.011	1.83	1.18	2.82	0.006
**Maternal Literacy**																
Cannot read at all					1.00				1.00				1.00			
Able to read only part of sentences					0.97	0.64	1.47	0.873	0.95	0.62	1.46	0.826	1.00	0.67	1.48	0.983
**Access to media**																
**Frequency of reading newspaper or magazine**																
Not at all									1.00							
Less than once a week									1.08	0.76	1.53	0.655	1.03	0.73	1.46	0.860
At least once a week/Almost every day									0.83	0.53	1.30	0.404	0.83	0.53	1.31	0.420
**Frequency of listening to Radio**																
Not at all									1.00							
Less than once a week									0.99	0.82	1.19	0.878	0.91	0.75	1.11	0.362
At least once a week/Almost every day									1.01	0.84	1.22	0.916	0.98	0.81	1.19	0.846
**Frequency of watching Television**																
Not at all									1.00							
Less than once a week									1.03	0.83	1.27	0.800	1.02	0.81	1.28	0.883
At least once a week/Almost every day									1.02	0.83	1.27	0.543	0.93	0.74	1.17	0.551
**Healthcare utilization factors**																
**Place of delivery**																
Home													1.00			
Health facility													1.09	0.84	1.43	0.508
**Mode of delivery**																
Non-caesarean													1.00			
Caesarean section													1.24	0.84	1.83	0.270
**Type of delivery assistance**																
Health professional													1.00			
Traditional birth attendant.													0.80	0.53	1.21	0.285
Other untrained													1.00	0.71	1.41	0.993
No one													0.81	0.59	1.10	0.178
**Antenatal Clinic visits**																
None													1.00			
1–3													1.52	1.14	2.03	0.004
≥4													1.66	1.24	2.23	0.001

^ Divorce/separated/widowed.

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
