# Peer review of "Exclusive Breastfeeding Rates and Associated Factors in 13 “Economic Community of West African States” (ECOWAS) Countries"

_nutrients, 2019, doi:10.3390/nu11123007_

Round 1
Reviewer 1 Report
Exclusive breastfeeding rates and associated factors in 13 Economic Community of West African States (ECOWAS) countries
This is an interesting paper that uncovers some aspects of breastfeeding on Western African countries; its impact from the public health point of view is strong.
Minor Comments:
The paper would benefit from some editing by an English native speaker.
Please change throughout the paper multivariate for multivariable
I was not able to find what * is in Table 2
Author Response
This is an interesting paper that uncovers some aspects of breastfeeding on Western African countries; its impact from the public health point of view is strong.
Response
Thank you for the comment.
Minor Comments:
The paper would benefit from some editing by an English native speaker.
Response
As noted above, the revised manuscript has been extensively and thoroughly edited by co-authors who are native English speakers.
Please change throughout the paper multivariate for multivariable
Response
Done
I was not able to find what * is in Table 2
Response:
* refers to the weighted sample size and percentages. We apologize for the omission.
Reviewer 2 Report
This is a really interesting analysis of DHS data and the key finding – that rates of exclusive breastfeeding in infants age 0-5 months is so low across these 13 countries - is hugely important. Strengths of this research include the use of DHS data, controlling for a range of relevant covariates in the analysis, and consideration of a number of factors which might be associated with mothers’ breastfeeding practices. However, I have some concerns about the manuscript in its present form.
I fully agree that there are some really interesting findings presented but, for me, the rationale for conducting these analyses needs to be stronger and presented more overtly (around lines 75-79). The research aims need to be clearly articulated so that it is clear by line 90 what this analysis aims to tell us, not just why such information might be useful. Line 102-105 – further details are required here so that it’s clear how and when these data were obtained, particularly the EBF information. Did mothers report this in an interview? If so, how old was the child when this was done? Understanding the details here is important for contextualising the results and understanding their potential reliability. Results section: The analyses conducted are extensive but they are somewhat hard to follow. I would suggest greater signposting to lead the reader through. Having specific aims, which are referred to in the results, will also aid interpretability of the array of findings. Additionally, not all of the results presented in the tables are described in the Results section or included in the Discussion. I feel that this is an oversight as important variables are not considered. I would question the need to present all of the results (e.g., all of the findings in Table 2) if they are not referred to in the text (Results and Discussion). Where results are not statistically significant is often just as interesting as where they are. Throughout: The manuscript was often hard to follow due to numerous errors with language use. I have not listed them all, but a really thorough proof-read is required to improve the clarity of messaging.
More specific/minor points
Line 48: define/explain “Economic Community of West African States” for readers unfamiliar with this term. Line 62 – why “will” this “predispose the infant to gastrointestinal infections…”. Ensure that all such points are fully explained and described. Line 75 – reword this as these factors are unlikely to be “attributable to EBF”. Rather, it is more likely that EBF is related to these factors. Line 84 – is “influencing” the right word here? It implies causality. Can this be determined? Line 82 – which “region”? Line 100 – “19735 infants” from how many? Line 101 – suggest rewording as “2.1 Outcome, confounding and exploratory variables” Line 111 – what does “place of residence” refer to? Can you give examples? Lines 121-124 – I suggest this information is augmented to make it clearer, for readers who are not familiar with this approach, exactly what was done and why. Line 147 – should this be 0-5 months (not 0-6)? Same on line 182. Line 158, Line 175 – check the subject of these sentences. I don’t think it should be the “Children” or the “Infants”. Lines 147-150 – I found this description of the findings rather hard to follow, possibly due to the fact that several concepts are covered in one sentence. Could this be reworded and clarified? Lines 165-167 – this very interesting finding warrants discussion. Is caesarean a proxy for wealth in many ECOWAS countries? Why might this result have occurred? Information about antenatal clinic visits seems to be presented in lines 167-170 and 177-180. While the numbers seem to differ, the overall findings seem the same (more EBF linked to more antenatal visits). Are both sets of information necessary? Line 220-221 – this suggestion needs to be further explained and, ideally, backed up with evidence. When comparing current findings to past studies, it would be really valuable to include the country for the previous research (e.g., lines 206, 214, 217, 228 etc). Line 237 – presumably this is due to mothers’ PERCEPTIONS of inadequate breast milk, not a verified drop in supply for mothers of boys? Line 261-262 – please explain what the “EBF investigation” was, when it occurred and how the 24-hour recall method was used. Line 269-270 – where is the evidence that mothers are offering water/other liquids to their child too early? If this is from past research, that research should be cited here.Author Response
Reviewer 2
Comments and Suggestions for Authors
This is a really interesting analysis of DHS data and the key finding – that rates of exclusive breastfeeding in infants age 0-5 months is so low across these 13 countries - is hugely important. Strengths of this research include the use of DHS data, controlling for a range of relevant covariates in the analysis, and consideration of a number of factors which might be associated with mothers’ breastfeeding practices. However, I have some concerns about the manuscript in its present form.
Response
Thank you for the comment. The reviewer concerns are addressed below in this rebuttal.
I fully agree that there are some really interesting findings presented but, for me, the rationale for conducting these analyses needs to be stronger and presented more overtly (around lines 75-79).
Response
As noted above and in response to the Academic Editor and reviewer #1, the revised manuscript (including the study rationale) has been extensively and thoroughly edited (Line 64-96).
The research aims need to be clearly articulated so that it is clear by line 90 what this analysis aims to tell us, not just why such information might be useful.
Response
Done (Line 93-96)
Line 102-105 – further details are required here so that it’s clear how and when these data were obtained, particularly the EBF information. Did mothers report this in an interview? If so, how old was the child when this was done? Understanding the details here is important for contextualising the results and understanding their potential reliability.
Response
The text has been clarified in the revised manuscript (Line 113-9)
Results section: The analyses conducted are extensive but they are somewhat hard to follow. I would suggest greater signposting to lead the reader through.
Response
The text has been clarified in the revised manuscript (139-59)
Having specific aims, which are referred to in the results, will also aid interpretability of the array of findings.
Response
We have clarified the text in the result section of the revised manuscript (Line 165-223)
Additionally, not all of the results presented in the tables are described in the Results section or included in the Discussion. I feel that this is an oversight as important variables are not considered.
Response
We thank the reviewer for the comment. We, however, note that the revised description of the results is consistent with research aim and reviewer #2 comment above (Line 165-223).
I would question the need to present all of the results (e.g., all of the findings in Table 2) if they are not referred to in the text (Results and Discussion).
Response
The results presented in the tables (e.g., all of the findings in Table 2) follows the conceptual model described in Table 1. This information is useful to potential readers to help contextualized our methodology and results. As noted above, the results have been extensively edited based on reviewer # 2 comment.
Where results are not statistically significant is often just as interesting as where they are.
Response
We appreciate the comment and note that the reviewer is correct. We are, however, guided by the word limit and succinctness and clarity of the manuscript narrative, as well as comments by reviewer # 2.
Throughout: The manuscript was often hard to follow due to numerous errors with language use. I have not listed them all, but a really thorough proof-read is required to improve the clarity of messaging.
Response
As noted above, the revised manuscript has been extensively edited.
More specific/minor points
Line 48: define/explain “Economic Community of West African States” for readers unfamiliar with this term.
Response
Done (Line 59-65)
Line 62 – why “will” this “predispose the infant to gastrointestinal infections…”. Ensure that all such points are fully explained and described.
Response
Done (Line 70-72)
Line 75 – reword this as these factors are unlikely to be “attributable to EBF”. Rather, it is more likely that EBF is related to these factors.
Response
Done (Line 75)
Line 84 – is “influencing” the right word here? It implies causality. Can this be determined?
Response
Done – line 87-96
Line 82 – which “region”?
Response
The text has been clarified in the revised manuscript – line 91
Line 100 – “19735 infants” from how many?
Response
The text has been clarified in the revised manuscript – line 111
Line 101 – suggest rewording as “2.1 Outcome, confounding and exploratory variables”
Response
Done – line 112
Line 111 – what does “place of residence” refer to? Can you give examples?
Response
Done – line 126
Lines 121-124 – I suggest this information is augmented to make it clearer, for readers who are not familiar with this approach, exactly what was done and why.
Response
The text has been clarified in the revised manuscript – line 144-150
Line 147 – should this be 0-5 months (not 0-6)? Same on line 182. Line 158, Line 175 – check the subject of these sentences. I don’t think it should be the “Children” or the “Infants”.
Response
We agree with the reviewer that “children” is an incorrect word in the manuscript. We have now clarified the text in the entire revised manuscript:
Lines 147-150 – I found this description of the findings rather hard to follow, possibly due to the fact that several concepts are covered in one sentence. Could this be reworded and clarified?
Response
We apologize that the reviewer found it hard to follow our description. The text has now been clarified in the revised manuscript – line 165-172
Lines 165-167 – this very interesting finding warrants discussion. Is caesarean a proxy for wealth in many ECOWAS countries? Why might this result have occurred?
Response
Additional text has been incorporated into the revised manuscript in response to the reviewer comment – line 303-310
We, however, note that caesarean section is not only affected by wealth but also by a number of other important factors, including place of residence, maternal education, pregnancy and childbirth complications (eclampsia, obstructed labour) etc.
Information about antenatal clinic visits seems to be presented in lines 167-170 and 177-180. While the numbers seem to differ, the overall findings seem the same (more EBF linked to more antenatal visits). Are both sets of information necessary?
Response
The reviewer sure has eyes for details – thank you for the observation! We originally presented results for univariate and multivariate analyses. The text in the univariate section has now been deleted in the revised manuscript.
Line 220-221 – this suggestion needs to be further explained and, ideally, backed up with evidence. When comparing current findings to past studies, it would be really valuable to include the country for the previous research (e.g., lines 206, 214, 217, 228 etc).
Response
Points appreciated and reflected in the entire revised manuscript.
Line 237 – presumably this is due to mothers’ PERCEPTIONS of inadequate breast milk, not a verified drop in supply for mothers of boys?
Response
The text has been clarified in the revised manuscript (line 286)
Line 261-262 – please explain what the “EBF investigation” was, when it occurred and how the 24-hour recall method was used.
Response
The text has been clarified in the revised manuscript (Line 321-2)
Line 269-270 – where is the evidence that mothers are offering water/other liquids to their child too early? If this is from past research, that research should be cited here.
Response
References have been incorporated as requested by the reviewer (Line 330)
Reviewer 3 Report
This is an interesting study that uses DHS data well to explore factors associated with exclusive breastfeeding in ECOWAS countries. However, the key issue with the paper, in my opinion, is how the results are presented and I believe this needs considerable reworking. I recommend that this paper make some minor revisions and I have several suggestions for improvement as detailed below. Please also make sure you check your wording and grammar to ensure your sentences make sense and your points are made effectively. I recommend that this paper is properly proofread again.
Throughout the paper there is inconsistency in how you refer to your sample. The age range of infants in your sample is described as 0-5 months in some places and 0-6 months in others. Please be clear which infants are included in your sample and be consistent with how this is described throughout. Please check the DHS breastfeeding questions, definitions and calculations carefully to make sure you are describing your sample appropriately. How do your calculations compare to the office DHS measure (see https://dhsprogram.com/Data/Guide-to-DHS-Statistics/Breastfeeding_and_Complementary_Feeding.htm)? This measure includes mother-infant dyads where the infant is 0-23 months and living with the mother – is this the same for your sample? In your abstract you present the range of EBF rates but you do not make it clear which age range this refers to. Consider revising to “EBF rates for infants 6 months or younger ranged from 13-58% in ECOWAS countries” (presuming this is the correct interpretation of the age band you have included). Please also describe your included sample more clearly (lines 99 and 100) as it currently sounds as though you are only including mother-infant dyads where the infant is 6 months or younger but in the discussion (line 259) you refer to delivery being as much as 5 years ago for some mothers. There is a discrepancy here and it is not clear whether mothers with older children are also included, but that they just provide answers for their youngest child. Your unit of analysis appears to be the mother-infant dyad rather than the mothers or the infants themselves, and as such care should be taken when describing associations, because in some cases variables relate to the mother and in others they relate to the infant. For example, it doesn’t make sense to say that “infants living in rural areas and female infants reported higher odds of EBF” (line 175) as the infants did not report these rates but rather their mothers did. Introduction: Please specify the target EBF required to reduce infant mortality and be specific about by how much infant mortality is expected to reduce (line 60). Regarding the studies describes in lines 72 to 75, it would be more informative to state the direction of association and to add the corresponding references after each factor as it is not currently clear which studies looked at what measures. Material and Methods: It would be helpful to define the terms birth rank and birth interval (lines 112) Please specify which statistical programme you used to conduct your analyse Results - Section 3.1: Line 143 should say “bivariate” rather than “univariate” or alternatively you can use “unadjusted” as you are looking at combinations of two variable, not just describing distributions of single variables Lines 147 – 149: if your point here is to illustrate the sample characteristics and highlight which countries the dyads came from, it may be more informative to say something like "The number of mother-infant dyads included in our sample varied by country, with Nigeria contributing the most (20% of our sample) and Togo the least (3.3%)". The comparison of EBF is a separate point and perhaps it would be more helpful to highlight the lowest and highest rates i.e. Cote d'Ivoire and Togo. The following point about infant size should also be made separately (lines 149-150) as currently it is confusing as it reads as if it is related to Nigerian dyads specifically, rather than the whole sample. Please reword lines 150-152 to make clearer and be consistent with your previous statement and report confidence intervals. You could also combine percentages to condense this point g. "mothers who perceived their babies to be average sized (43%) were 1.21 times as likely to EBF than those who perceived their infants to be small (19%) (95% CI 1.02, 1.42)". Please include indications of effect size (ie report odds ratios) for all associations described in lines 158-170. Please be consistent in the number of decimal points you report for your percentages Results Section 3.2: Line 172 – “Table 3 illustrates factors associated with exclusive breastfeeding…” this is what Table 2 shows, you need to make the distinction clearer i.e. that Table 23 shows adjusted associations and perhaps reiterate the model progression steps Separate out age and country information in lines 173 and 174 because otherwise it sounds as if you are referring to the effect of baby's age only within Nigeria and Cote d'Ivoire. I suggest that section 3.2 is rewritten to correspond to the model progression steps represented in the table and to be consistent with your justification of this approach (lines 137 and 138) i.e. highlight which (sets of) factors remained significant predictors throughout all model progression steps and which no longer became significant after adjusting for other factors, and how effect sizes changed. Please also make sure you write these results with the correct interpretation for example, being clear what is adjusted for when you present an odds ratio e.g. "mother's age significantly positively predicted exclusive breastfeeding; 35-49 year old mothers were 1.5 times more likely to EBF than those aged 15-19 years when just controlling for demographic factors and this effect persisted after additionally controlling for socioeconomic, media and healthcare factors (Model 4 OR 1.5, 95% CI 1.12-1.99)." I suggest picking up on the country differences more in your write up, e.g. how the difference between Mali's and Benin's EBF disappears throughout model progression, suggesting that socioeconomic factors explain the differences between the rates in the two countries. It is probably also helpful to point out that most of the between country differences in EBF persisted even after controlling for socioeconomic, media and healthcare factors and providing some discussion (in the discussion section) as to why this might be i.e. what other factors have you not measured that could account for these differences? The association between rural residence and EBF only emerges once socioeconomic factors are controlled for, and this needs to be made clear in your description. Make sure to report effect sizes (e.g. in reference to schooling associations (line 177)) and be clear about what is adjusted for when reporting associations (e.g. in reference to schooling associations (line 177) and ANC visits lines (177-180)). Figure 1: The Togo bar shouldn’t have a thicker outline - it might make more sense for the pooled estimate to be with a thicker outline to differentiate it from single country estimates Table 2: Not clear what the asterisks denote, please include in caption Should there be a row above country stating demographic factors in keeping with how you have presented results in Table 3? Table 3: Change Mail to Mali Correct table formatting so line is under demographic factors not Benin Discussion: Lines 185-186: EBF being strongly associated with rural residence may be too strong a claim given that rural residence only predicted EBF once socioeconomic factors were controlled for. Also direction of association is not clear in how you have worded this here. Line 188: Consider using "were only weakly associated with EBF" rather than reported weak relationship. You need to be clear about direction of association. And what do you mean by weak relationship? There is strong evidence for a gender effect (with low p-values) and a final OR of 1.17 shows a relatively small (rather than weak) effect. Be careful with wording. Lines 193-195: You control for whether mothers were working....you need to consider this in your interpretation. Please provide supporting references for the maternity leave provisions (line 199) and maternal age mechanisms (lines 220 and 221). Revise lines 201 to 203 as your results suggest that rural women are more likely to breastfeed. Be careful not to contradict yourself, and keep your argument consistent Please provide additional explanatory detail in lines 236 to 239 as it is not clear why mothers don’t think that breastmilk is inadequate for female demands too or what you are saying EBF is protective against. Conclusion: What were the levels expected to be (line 264)?

Author Response
Reviewer 3
This is an interesting study that uses DHS data well to explore factors associated with exclusive breastfeeding in ECOWAS countries. However, the key issue with the paper, in my opinion, is how the results are presented and I believe this needs considerable reworking. I recommend that this paper make some minor revisions and I have several suggestions for improvement as detailed below. Please also make sure you check your wording and grammar to ensure your sentences make sense and your points are made effectively. I recommend that this paper is properly proofread again.
Response
We thank the reviewer for the comment. The reviewer’s concerns are addressed below. We also note that the revised manuscript has been extensively edited for wording and syntax.
Throughout the paper there is inconsistency in how you refer to your sample. The age range of infants in your sample is described as 0-5 months in some places and 0-6 months in others. Please be clear which infants are included in your sample and be consistent with how this is described throughout. Please check the DHS breastfeeding questions, definitions and calculations carefully to make sure you are describing your sample appropriately. How do your calculations compare to the office DHS measure (see https://dhsprogram.com/Data/Guide-to-DHS-Statistics/Breastfeeding_and_Complementary_Feeding.htm)? This measure includes mother-infant dyads where the infant is 0-23 months and living with the mother – is this the same for your sample?
Response
The reviewer has eyes for specifics – thank you for the observation. The age range (0–6 months) was written in error and has now been edited in the entire revised manuscript to (0–5 months). We would like to reassure the reviewer that we used the appropriate Measure DHS definition for EBF as noted in the revised manuscript (Line 113-220). Our previously published studies on infant and young child feeding are noted below for the reviewer/editor consideration:
Agho, K., Dibley, M., Odiase, J., & Ogbonmwan, S. (2011). Determinants of exclusive breastfeeding in Nigeria. BMC pregnancy and childbirth, 11, 2. Agho, K. E., Ogeleka, P., Ogbo, F. A., Ezeh, O. K., Eastwood, J., & Page, A. (2016). Trends and predictors of prelacteal feeding practices in Nigeria (2003–2013). Nutrients, 8(8), 462. Ogbo, F. A., Agho, K., Ogeleka, P., Woolfenden, S., Page, A., & Eastwood, J. (2017). Infant feeding practices and diarrhoea in sub-Saharan African countries with high diarrhoea mortality. Plos One, 12(2), e0171792. Ogbo, F. A., Eastwood, J., Page, A., Efe-Aluta, O., Anago-Amanze, C., Kadiri, E. A., . . . Agho, K. E. (2017). The impact of sociodemographic and health-service factors on breast-feeding in sub-Saharan African countries with high diarrhoea mortality. Public health nutrition, 20(17), 3109-3119. Ogbo, F. A., Page, A., Idoko, J., Claudio, F., & Agho, K. E. (2015). Trends in complementary feeding indicators in Nigeria, 2003–2013. BMJ Open, 5(10), e008467. Ogbo, F. A., Dhami, M. V., Awosemo, A. O., Olusanya, B. O., Olusanya, J., Osuagwu, U. L., . . . Agho, K. E. (2019). Regional prevalence and determinants of exclusive breastfeeding in India. International breastfeeding journal, 14, 20. Ogbo, F. A., Ezeh, O. K., Khanlari, S., Naz, S., Senanayake, P., Ahmed, K. Y., . . . Page, A. (2019). Determinants of exclusive breastfeeding cessation in the early postnatal period among culturally and linguistically diverse (CALD) Australian mothers. Nutrients, 11(7), 1611.
In your abstract you present the range of EBF rates but you do not make it clear which age range this refers to. Consider revising to “EBF rates for infants 6 months or younger ranged from 13-58% in ECOWAS countries” (presuming this is the correct interpretation of the age band you have included).
Response:
Thank you for the observation. Yes, the reviewer interpretation is correct. We have now revised the text in the revised manuscript (abstract section)
Please also describe your included sample more clearly (lines 99 and 100) as it currently sounds as though you are only including mother-infant dyads where the infant is 6 months or younger but in the discussion (line 259) you refer to delivery being as much as 5 years ago for some mothers. There is a discrepancy here and it is not clear whether mothers with older children are also included, but that they just provide answers for their youngest child.
Response
The text has been clarified in the revised manuscript (Line 110)
Your unit of analysis appears to be the mother-infant dyad rather than the mothers or the infants themselves, and as such care should be taken when describing associations, because in some cases variables relate to the mother and in others they relate to the infant. For example, it doesn’t make sense to say that “infants living in rural areas and female infants reported higher odds of EBF” (line 175) as the infants did not report these rates but rather their mothers did.
Response
The text has been clarified in the entire revised manuscript.
Introduction: Please specify the target EBF required to reduce infant mortality and be specific about by how much infant mortality is expected to reduce (line 60).
Response
Done (Line 68):
Regarding the studies describes in lines 72 to 75, it would be more informative to state the direction of association and to add the corresponding references after each factor as it is not currently clear which studies looked at what measures.
Response
Done (Line 76-86):
Material and Methods: It would be helpful to define the terms birth rank and birth interval (lines 112)
Response
Done (Page 126-130)
Please specify which statistical programme you used to conduct your analyse
Response
We apologize for this omission. The text has been incorporated into the entire revised manuscript (Line 158-160):
Results - Section 3.1: Line 143 should say “bivariate” rather than “univariate” or alternatively you can use “unadjusted” as you are looking at combinations of two variable, not just describing distributions of single variables
Response
Done
Lines 147 – 149: if your point here is to illustrate the sample characteristics and highlight which countries the dyads came from, it may be more informative to say something like "The number of mother-infant dyads included in our sample varied by country, with Nigeria contributing the most (20% of our sample) and Togo the least (3.3%)".
Response
Done (Line 165-72)
The comparison of EBF is a separate point and perhaps it would be more helpful to highlight the lowest and highest rates i.e. Cote d'Ivoire and Togo.
Response
The text has been clarified in the revised manuscript (Line 165-72)
The following point about infant size should also be made separately (lines 149-150) as currently it is confusing as it reads as if it is related to Nigerian dyads specifically, rather than the whole sample.
Response
This text was deleted from the revised manuscript based on the reviewer #2 comment above.
Please reword lines 150-152 to make clearer and be consistent with your previous statement and report confidence intervals. You could also combine percentages to condense this point g. "mothers who perceived their babies to be average sized (43%) were 1.21 times as likely to EBF than those who perceived their infants to be small (19%) (95% CI 1.02, 1.42)".
Response
As noted above, this text was deleted from the revised manuscript based on the reviewer #2 comment above.
Please include indications of effect size (ie report odds ratios) for all associations described in lines 158-170.
Response
Done (Line 180-96)
Please be consistent in the number of decimal points you report for your percentages
Response
Points appreciated and reflected in the entire revised manuscript.
Results Section 3.2: Line 172 – “Table 3 illustrates factors associated with exclusive breastfeeding…” this is what Table 2 shows, you need to make the distinction clearer i.e. that Table 23 shows adjusted associations and perhaps reiterate the model progression steps Separate out age and country information in lines 173 and 174 because otherwise it sounds as if you are referring to the effect of baby's age only within Nigeria and Cote d'Ivoire.
Response
Done (Line 180-96)
I suggest that section 3.2 is rewritten to correspond to the model progression steps represented in the table and to be consistent with your justification of this approach (lines 137 and 138) i.e. highlight which (sets of) factors remained significant predictors throughout all model progression steps and which no longer became significant after adjusting for other factors, and how effect sizes changed.
Response
The text has been clarified in the revised manuscript (Line 198-223). We note, however, that highlighting (i.e., rewriting) the information that is already in the methods and tables as to which factors remained significant in the final model may be confusing for some potential readers in an already ‘busy or lengthy’ results section. We have presented the information in the tables for interested readers to consider. Based on the initial presentation of the results, reviewer # 2 raised concerns that we should delete some of the data from the tables.
We believe that the revised presentation of the results is succinct and easy to follow by interested readers. Nevertheless, we would like to defer the final decision on the comment to the Academic Editor.
Please also make sure you write these results with the correct interpretation for example, being clear what is adjusted for when you present an odds ratio e.g. "mother's age significantly positively predicted exclusive breastfeeding; 35-49 year old mothers were 1.5 times more likely to EBF than those aged 15-19 years when just controlling for demographic factors and this effect persisted after additionally controlling for socioeconomic, media and healthcare factors (Model 4 OR 1.5, 95% CI 1.12-1.99)."
Response
As noted above, we would like to defer the final decision on this suggestion to the Editor. However, we note that the phrase ‘multivariate’ in epidemiology and biostatistics (as stated in the methods and results sections of the manuscript) means adjustment for covariates. So, the addition of more text around adjustments for covariates in all data interpretation would make the reading of the results somewhat uninteresting for the reader, especially that the information is already in the tables.
I suggest picking up on the country differences more in your write up, e.g. how the difference between Mali's and Benin's EBF disappears throughout model progression, suggesting that socioeconomic factors explain the differences between the rates in the two countries.
Response
As above, this information is in the tables for interested readers to consider. The manuscript is guided by the focused and succinct research question (i.e., what is the role of sociodemographic, health, media and country-level factors on EBF in 13 ECOWAS countries?). We note that the assessment of between-country comparisons alongside the impact of sociodemographic among other factors may be out of scope for the current manuscript and may need further consideration.
We thank the reviewer for this suggestion as we are now considering examining this research question: what is the variation and impact of socioeconomic inequalities on infant and young feeding practices across ECOWAS countries?
It is probably also helpful to point out that most of the between country differences in EBF persisted even after controlling for socioeconomic, media and healthcare factors and providing some discussion (in the discussion section) as to why this might be i.e. what other factors have you not measured that could account for these differences?
Response
Done (Line 206-8). We noted that information relating to unmeasured covariates has now been noted in the limitation section of the manuscript (line 318-20).
The association between rural residence and EBF only emerges once socioeconomic factors are controlled for, and this needs to be made clear in your description.
Response
Done (Lie 215-6)
Make sure to report effect sizes (e.g. in reference to schooling associations (line 177)) and be clear about what is adjusted for when reporting associations (e.g. in reference to schooling associations (line 177) and ANC visits lines (177-180)).
Response
Done (Line 198-223)
Figure 1: The Togo bar shouldn’t have a thicker outline - it might make more sense for the pooled estimate to be with a thicker outline to differentiate it from single country estimates
Response
Agreed and we have adjusted the figure.
Table 2: Not clear what the asterisks denote, please include in caption Should there be a row above country stating demographic factors in keeping with how you have presented results in Table 3?
Response
Thank you for the observation – done.
Table 3: Change Mail to Mali Correct table formatting so line is under demographic factors not Benin
Response
Thank you for the observation – done.
Discussion: Lines 185-186: EBF being strongly associated with rural residence may be too strong a claim given that rural residence only predicted EBF once socioeconomic factors were controlled for.
Response
Text now edited (Line 225-231)
Also direction of association is not clear in how you have worded this here.
Response
Text now edited (Line 225-231)
Line 188: Consider using "were only weakly associated with EBF" rather than reported weak relationship.
Response
Text now edited (Line 225-231)
You need to be clear about direction of association. And what do you mean by weak relationship? There is strong evidence for a gender effect (with low p-values) and a final OR of 1.17 shows a relatively small (rather than weak) effect. Be careful with wording.
Response
Points appreciated and reflected in the revised manuscript.
Lines 193-195: You control for whether mothers were working....you need to consider this in your interpretation.
Response
Points appreciated and now reflected in the revised manuscript.
Please provide supporting references for the maternity leave provisions (line 199) and maternal age mechanisms (lines 220 and 221).
Response
Done (Line 243-45):
Revise lines 201 to 203 as your results suggest that rural women are more likely to breastfeed.
Response
Points appreciated and now reflected in the revised manuscript (Line 243-50)
Be careful not to contradict yourself, and keep your argument consistent Please provide additional explanatory detail in lines 236 to 239 as it is not clear why mothers don’t think that breastmilk is inadequate for female demands too or what you are saying EBF is protective against.
Response
Points appreciated and now reflected in the revised manuscript (line 234-40)
Conclusion: What were the levels expected to be (line 264)?
Response
Points appreciated and now reflected in the revised manuscript (Line 325)